# All-Trans Retinoic Acid Fosters the Multifarious U87MG Cell Line as a Model of Glioblastoma

**DOI:** 10.3390/brainsci11060812

**Published:** 2021-06-18

**Authors:** Markéta Pokorná, Michael Hudec, Iva Juříčková, Michael Vácha, Zdeňka Polívková, Viera Kútna, Jan Pala, Saak V. Ovsepian, Marie Černá, Valerie Bríd O’Leary

**Affiliations:** 1Department of Medical Genetics, Third Faculty of Medicine, Charles University, Ruská 87, Vinohrady 10000 Prague, Czech Republic; Marketa.pokorna@icloud.com (M.P.); Michael.hudec@lf3.cuni.cz (M.H.); iva.jurickova@lf3.cuni.cz (I.J.); michael.vacha@seznam.cz (M.V.); zdenka.polivkova@lf3.cuni.cz (Z.P.); jan.pala@lf3.cuni.cz (J.P.); marie.cerna@lf3.cuni.cz (M.Č.); 2Department of Experimental Neurobiology, National Institute of Mental Health, Topolová 748, 250 67 Klecany, Czech Republic; viera.kutna@nudz.cz (V.K.); saak.ovsepian@nudz.cz (S.V.O.)

**Keywords:** chromosome, lncRNA, CD54, prominin-1, ATRA, brain cancer

## Abstract

Glioblastoma multiforme (GBM) is a primary brain cancer of poor prognosis, with existing treatments remaining essentially palliative. Current GBM therapy fails due to rapid reappearance of the heterogeneous neoplasm, with models suggesting that the recurrent growth is from treatment-resistant glioblastoma stem-like cells (GSCs). Whether GSCs depend on survival/proliferative cues from their surrounding microenvironmental niche, particularly surrounding the leading edge after treatment remains unknown. Simulating human GBM in the laboratory relies on representative cell lines and xenograft models for translational medicine. Due to U87MG source discrepancy and differential proliferation responses to retinoic acid treatment, this study highlights the challenges faced by laboratory scientists working with this representative GBM cell line. Investigating the response to all trans-retinoic acid (ATRA) revealed its sequestering of the prominin-1 stem cell marker. ICAM-1 universally present throughout U87MG was enhanced by ATRA, of interest for chemotherapy targeting studies. ATRA triggered diverse expression patterns of long non-coding RNAs *PARTICLE* and *GAS5* in the leading edge and established monolayer growth zone microenvironment. Karyotyping confirmed the female origin of U87MG sourced from Europe. Passaging U87MG revealed the presence of chromosomal anomalies reflective of structural genomic alterations in this glioblastoma cell line. All evidence considered, this study exposes further phenotypic nuances of U87MG which may belie researchers seeking data contributing towards the elusive cure for GBM.

## 1. Introduction

Transformation of benign astrocyte glial cells instigates the development of fast-growing infiltrative glioblastoma multiforme (GBM), predominantly within the human cerebral cortex [1]. Such proliferating brain cancer activates adjacent normal stromal tissue via the release of pro-tumorigenic factors into their surrounding microenvironment [2]. A dearth of molecular tests has been recognized as an obstacle for evolving clinical practices related to GBM treatment which to date remains non-curative. Scientific evidence is heavily reliant on in vitro cultured cell lines (e.g., human-derived U251, U87, murine GL261 and rat 9L/LacZ, F98, RG2, CNS-1, and C6) [3] and xenografted animal models, to simulate human GBM for translational medicine yet clinical trials have still proven ineffective [4]. Variance exists when attempting to utilize cell line data and animal models due to differences between the in vivo microenvironment and the artificial conditions generated in vitro when growing cells in suspension or in adhesion to plastic surfaces. Appropriate and predictive preclinical animal glioblastoma models generated from xenografted glioma cell lines require optimized implantation sites for assisting the development of innovative GBM therapies [3,5,6].

Therapeutic retinoids operating on a feedback inhibition basis [7], recently sparked a promising GBM treatment strategy utilizing in vitro poly (diol citrate) wafers to ensure the gradual release of all-trans retinoic acid (ATRA) [8]. Cultured astrocytes have been shown to actively synthesize ATRA, the most biologically active metabolite of vitamin A [9], for potential neuronal differentiation [10]. Data revealed that U87MG (Uppsala 87 Malignant Glioma) differentiation levels varied depending on whether ATRA was released from a polymeric wafer or added directly to culture media [8]. Contingent to retinoic acid concentration, opposite proliferation effects on human glioblastoma cell lines were noted [8].

The human derived cell line U87MG was originally obtained from a 44-year-old female patient in 1966 at Uppsala University, Sweden. This cell line has been a hallmark of scientific investigation resulting in thousands of publications offering contributions to glioblastoma research [11]. A comparison of gene-expression profiles between the U87MG cell line distributed by American Type Culture Collection (ATCC) cell repository and the original tumor tissue began after a higher proliferation rate was noted in the former as well as the identification of a Y chromosome [12]. In recognition of the discrepancy, ATCC currently marks U87MG with the identifier HTB-14^TM^ and the description that the cell line is likely to be a glioblastoma of male CNS origin [12].

Operated by Public Health England, U87MG is also available from the European Collection of Authenticated Cell Cultures (ECACC). They indicate that U87MG is derived from a female patient with malignant glioma. This study aimed to investigate the proliferative characteristics, gene expression profile and karyotype of U87MG (ECACC 89081402) post ATRA exposure to highlight subtle challenges associated with this widely utilized laboratory cell line. Given the discrepancy that evidently occurred previously between suppliers and research institutes, it was deemed prudent to characterize U87MG (ECACC 89081402) given recent calls by the neuro-oncology community for the relegation of this cell line as a prerequisite for more representative GBM models.

## 2. Materials and Methods

### 2.1. Propagation and Maintenance of U87MG

U87MG (ECACC; cat. #89081402) was grown in Dulbecco modified eagle medium (DMEM; 4.5 g/L D-glucose, 4 μM L-glutamine and 1 mM sodium pyruvate additive), fetal bovine serum (FBS) (10%; Life Technologies, Prague, Czech Republic), penicillin and streptomycin (1%; Sigma-Aldrich^®^, St. Louis, MO, USA). Cultures were routinely checked for mycoplasma contamination using a LookOut Mycoplasma PCR Detection Kit (Sigma Aldrich, St. Louis, MO, USA, cat. #IMP0035-1KT). In general, cells were grown (with media change every 3 days) in a humidified incubator with 5% CO_2_ at 37 °C, to 80% confluency prior to removal from the dish using trypsin (0.25%)/EDTA (0.02%) and sub-culturing (passaging) or harvesting.

### 2.2. All Trans-Retinoic Acid (ATRA) Preparation

As ATRA (cat. #44540.77, Thermo-Fisher Scientific, Waltham, MA, USA) is extremely sensitive to UV light and air, the entire contents of the purchased ampoule were dissolved in dimethyl sulfoxide as a stock solution. A stock (10 mM) was dissolved in DMEM for a final concentration of 100 μM. U87MG were exposed to an ATRA range of 10^−7^ M–10^−4^ M for up to 7 days.

### 2.3. Wound Healing Assay

U87MG were seeded at 1 × 10^5^ cells/well and cultivated in 6-well dishes to 80% confluency. Cells were removed with a scraper to create a gap from a demarcated central 1 cm zone in the well and washed twice with PBS. Media was changed to include a range of ATRA concentrations (0–10^−4^ M) per dish for up to 7 days when wound closure was complete. The central or peripheral zones of the dish represented the leading edge (LE) and established cellular monolayer (ECM) of the U87MG culture. Cellular proliferation into the previously cleared region (LE) and within the ECM was monitored over 7 days using an upright light microscope (Nikon Eclipse TS100, Nikon, Tokyo, Japan) equipped with mounted camera (Canon, Tokyo, Japan) and finally removed for total RNA extraction, gene expression assessment or flow cytometry.

### 2.4. Determination of Proliferation with EdU Flow Cytometry

A Click-iT^TM^ Plus EdU flow cytometry assay kit (Thermo-Fisher Scientific, Waltham, MA, USA cat #C10634) was utilized to measure U87MG proliferation in the LE and ECM zones of the culture exposed to ATRA concentrations (0–10^−4^ M) as outlined above. EdU (5-ethynyl-2′-deoxyuridine) is a nucleoside analog of thymidine which is incorporated into DNA during active DNA synthesis. The detection is based on a copper catalyzed covalent reaction between picolyl azide coupled to Alexa Fluor^TM^ 647 dye and an alkyne provided by the ethyl moiety of the EdU. The procedure was followed as per manufacturer’s instructions with EdU (1 µM per well) exposure for 48 h during wound closure. Cells were removed from the LE and ECM zones with a cell scraper which was rinsed in 1% BSA in PBS. Cells were pelleted by centrifugation at 4000× *g* and resuspended in 4% paraformaldehyde in PBS (100 µL). Following incubation for 15 min in the dark, cells were washed in 1% BSA in PBS (3 mL) and pelleted by centrifugation at 4000× *g*. Cells were resuspended in a saponin-based permeabilization and wash reagent (1×) with incubation for 15 min. A Click-iT Plus reaction cocktail (500 µL per sample) was prepared according to instructions which included PBS, copper protectant, Alexa Fluor 647 picolyl azide and a buffer additive. The reaction mixture was incubated for 30 min at RT with protection from the light. Cells were washed in saponin-based permeabilization and wash reagent (1×) and transferred to polystyrene round bottom tubes (cat. #352054, Corning, New York, NY, USA) for analysis on a BD FACSVerse flow cytometer. Sample measurements were taken using allophycocyanin channel settings (Ex-Max: 650 nm; Em-Max: 660 nm) due to the near identical spectra to that of Alexa Fluor 647 picolyl azide. Fluorescence intensity, cell counts and side scatter profiles were determined through cloud-based capabilities and histogram determination.

### 2.5. Total RNA Isolation and cDNA Synthesis

Total RNA was isolated from U87MG using a MirVana^TM^ miRNA Isolation Kit (Ambion RNA Life Technologies, Prague, Czech Republic, cat. #AM1560). It should be noted that this protocol extracts total RNA with the option for further purification of RNA enriched for small RNAs which was not done for this study. In brief, 1 × 10^6^ cells were disrupted using lysis solution, with RNA extracted via phenol/chloroform and ethanol precipitation. RNA purification was carried out with solid phase filter cartridges using appropriate washing solutions with final elution in 95 °C ultrapure water. Concentration and purity assessment of total RNA was achieved using O.D. 260/280 ratio determination (NanoDrop 1000, Thermo-Fisher Scientific, Waltham, MA, USA). Total RNA was stored at −80 °C. Total RNA (100 ng) from ±ATRA exposed U87MG cells was converted into first strand cDNA using standard protocol procedures (with the inclusion of random hexamers) and reagents (Thermo Fisher Scientific, cat. #18091050).

### 2.6. Real Time PCR Quantification of Long Non-Coding RNA and Endogenous Control Genes

Pre-designed single Taqman gene expression assays were purchased (Thermo-Fisher Scientific, Waltham, MA, USA) for the assessment of long non-coding RNAs (*PARTICLE* [13,14,15,16,17] and *GAS5* [18,19,20]) and GAPDH as an endogenous control gene. The reaction conditions for single gene assays were as reported [15,20]. In brief: cDNA (50–100 ng), 1× Taqman universal PCR master mix (no AmpErase UNG; Life Technologies, Prague, Czech Republic, cat. #4324018), forward and reverse primers (10 pmol), specific fluorescent probe (5 pmol), nuclease-free water up to 25 µL. Negative controls were represented as samples with the absence of template. StepOne™ Real-time PCR systems (Life Technologies, Prague, Czech Republic) enabled holding (50 °C, 2 min; 95 °C, 10 min) and cycling (95 °C, 15 s; 60 °C 1 min; 40 cycles). Cycle threshold values were extracted and fold changes in gene expression determined by 2^(−∆∆Ct)^. Naïve samples (non ATRA exposed) were normalized to a value = 1 with test samples relatively compared.

### 2.7. Protein Extraction

Protein extraction from U87MG cells was performed using a T-PER reagent (cat. #78510, Thermo-FisherScientific, Waltham, MA, USA) with addition of protease inhibitor cocktail tablets (cat. #04693116001, Roche, Basel, Switzerland). 100 μL of the reagent was added to the 1 × 10^6^ cells followed by homogenization and sonication (20 s) for membrane disruption. Cells were centrifuged for 5 min at 10,000× *g* to remove cell debris. Protein concentration was determined using a bicinchoninic acid (BCA) assay (cat. #23227, Thermo-FisherScientific, Waltham, MA, USA).

### 2.8. Electrophoresis and Western Blotting

Cell lysates (25 µg, 10 μL) were mixed with 4× NuPage LDS Sample Buffer (Life Technologies, Prague, Czech Republic) (2.5 μL) and heated for 5 min at 70 °C before loading onto 12% Bis Tris NuPage gels (cat. #NP0342BOX, Novex Life Technologies, Prague, Czech Republic) with electrophoresis in 1× MOPS—SDS running buffer at 180 V in 4 °C. Separated proteins were transferred onto Nytran membranes under standard conditions followed by blocking in 5% bovine serum albumin in TBS-T. Detection of prominin 1/CD133 or ICAM-1/CD54 were determined with over-night incubation at 4 °C (1:1000 in blocking reagent) using rabbit monoclonal anti-prominin 1/CD133 (cat. #ab216323, abcam, Cambridge, United Kingdom,) or rabbit monoclonal anti-CD54/ICAM-1 (cat. #4915, Cell Signaling Technology, Danvers, MA, USA) respectively. GAPDH was also detected as a comparative control using a mouse anti-GAPDH mAb (cat. #C2514, Santa Cruz Biotechnology, Dallas, TX, USA). Following extensive washing with TBS-T, Nytran membranes were exposed to alkaline phosphatase-conjugated goat anti-rabbit (1:5000) (cat. #A-3687, Sigma-Aldrich^®^, St. Louis, MO, USA) or goat-anti-mouse (1:5000) (cat. #A-3562, Sigma-Aldrich^®^, St. Louis, MO, USA) secondary antibody for 1 h at RT. Specific proteins (Prominin1/CD133, ICAM1/CD54 and GAPDH) were visualized using a mix of 5-bromo-4-chloro-3′-indolyl-phospate p-toluidine and nitro-blue tetrazolium chloride solution (cat. #1001973039, Sigma-Aldrich^®^, St. Louis, MO, USA). Western blots were photographed using a FluorChem HD2 gel visualization system (Alpha Innotec, Kasendorf, Germany) with specific protein band intensities quantified using ImageJ (NIH, Bethesda, Rockville, MD, USA).

### 2.9. Cytogenetic Harvesting of U87MG

Cells were grown to 60–70% confluence with exposure to ATRA (10^−4^ M), washed in phosphate buffered saline and exposed for 17 h to growth medium (outlined above) containing colchicine (250 µg/mL) in standard incubation conditions [21]. Cells were removed with trypsin/EDTA and treated with warm (37 °C) hypotonic solution (ultra-pure water and growth medium, 3:1 ratio respectively) for 20 min at RT. Following centrifugation (1000× *g* for 15 min), the pellet was gently resuspended in ice cold fixative (10 mL; methanol: glacial acetic acid, 3:1 ratio respectively). Centrifugation was repeated as previously done and pellet resuspended as before but in a reduced volume of ice-cold fixative (2 mL) followed by incubation at 4 °C for 20 min. After a final centrifugation (1000× *g* for 15 min), the chromosomal pellet was resuspended in acetic acid (200 µL). Ultra-clean microscope slides were prepared with overnight treatment in hot (65 °C) concentrated HCl, followed by three rounds of sonication in sterile distilled water. Cell suspension solution (25 µL) was applied from a height to cold wet ultra-clean microscope slides and dried on a hotplate (50 °C) for 15 min.

### 2.10. Wright Staining and Banding of U87MG Chromosomes

Following cell suspension attachment, microscope slides were heated to 95 °C for 25 min. Then slides were immersed for 20 s in Sörensen phosphate buffer (SPB) containing trypsin (2 µg/mL; pH 6.8). The action of trypsin was stopped with subsequent immersion of slides for 30 s into SPB containing fetal bovine serum (10%; pH 6.8), rinsed in SPB and allowed to air dry. Wright staining solution (WSS: Hydrion buffer capsules diluted in dH_2_0, pH 6.8 (cat. #60784–226, Micro Essential Laboratory, New York, NY, USA) and Wright stain, 3:1 ratio) was prepared. The slides were placed in a horizonal position on a staining rack and entirely covered with WSS for 5 min. After washing with distilled water, slides were drained and allowed to air dry. At least 10 metaphase spreads of chromosomes per passage were viewed under an Olympus BX43F light microscope at 1000× magnification.

### 2.11. Giemsa Staining for Structural Chromosomal Abberations

Microscopic slides of U87MG cell suspension were heated as outlined above, allowed to come to RT and immersed in a vertical position into a coplin jar containing Giemsa staining solution (GSS: Giemsa stain in Sörensen phosphate buffer, pH 6.8, 1 in 20 dilution) for 5 min. Slides were transfered to distilled water for 2–3 s to remove excess stain and air dried. At least 10 metaphase spreads of chromosomes per passage were viewed under an Olympus BX43F light microscope at 1000× magnification with images obtained via green pseudo-coloration.

### 2.12. Immunofluorescence

U87MG were cultivated as previously described above on glass coverslips. Having reached ~80% confluence, the media was removed and the cells were washed two times for 5 min with PBS. Cellular fixation occurred upon exposure to 4% paraformaldehyde for 1 h, followed by washing for 5 min with PBS. Cells were permeabilized in 1× TBST (1× TBS including 0.5% Triton™ X-100 (cat. #X100–5ML, Sigma-Aldrich^®^, St. Louis, MO, USA)) for 1 h. Following one wash for 5 min in 1× TBS, U87MG were placed in blocking solution (1× TBS containing 5% bovine serum albumin and 0.5% Triton™ X-100) for 1 h at room temperature (RT). Cells were then exposed to antibody representing rabbit monoclonal anti-prominin 1/CD133 (cat. #ab216323, abcam, Cambridge, United Kingdom, 1:200 in blocking solution) or rabbit monoclonal anti-CD54/ICAM-1 (cat. #4915, Cell Signaling Technology, New York, NY, USA, 1:200 in blocking solution) with o/n incubation at 4 °C. Samples were washed three times for 15 min in 1× TBST and incubated in Alexa fluor^®^ 488 goat anti-rabbit IgG (H + L) (1:500; in blocking solution) for 1 h at RT in the dark. This was followed by washing three times for 15 min with 1× TBST and air drying in the dark. To prepare for microscopy, cells on coverslips were mounted in ProLong^®^ Gold Antifade reagent (cat. #8961S, Cell Signaling Technology, New York, NY, USA) containing DAPI, and placed on a glass slide. The images were acquired with a Leica TCS SP8X confocal system (Leica Microsystems, Mannheim, Germany) using an HCX PL APO 40×/1.30 Oil objective and appropriate excitation 405 and 488 nm lasers, which were analyzed in LAS AF software (Leica Microsystems, Mannheim, Germany), and ImageJ 1.47 software (Bethesda, NIH). The brightness and contrast of images have been adjusted in a standardized manner for all of the images as described [22]. All images and graphs were generated and assembled in figures using Adobe Illustrator (Adobe Systems, San Jose, CA, USA).

### 2.13. Statistical Analysis

Two tailed Student’s t-test was employed for comparative purposes between groups. Data is presented as mean ± standard error with significance determined at *p* ≤ 0.05. Experiments were replicated *n* = 3.

### 2.14. Data Availability

All material and data will be available upon request.

## 3. Results

### 3.1. Reduced Expression of the Stem Cell Marker Prominin-1 (CD133) in U87MG in Response to ATRA

The prognostic value and distribution of prominin 1/CD133 in U87MG has proven to be controversial due to reported inconsistencies in the literature [23]. Availing of recently generated, more specific anti-CD133 antibodies, our results support the widespread distribution (92.4 ± 3%, CD133+ cells per field of view (FoV = 20)) of this protein within the cytoplasm of U87MG but not within the perinuclear space (Figure 1, Appendix A).

Prominin 1/CD133 revealed an inverse expression response to increased ATRA concentrations after 7 days exposure as shown by Western blotting analysis. Prominin 1/CD133 significantly decreased in U87MG exposed to ATRA (10^−6^ M, 1-fold, *p* = 0.003; 10^−4^ M, 1.7-fold, *p* = 0.000024) in comparison to non-treated cells (Figure 1). This data supports the role of ATRA in the inhibition of stem cell characteristics as seen with other cancer types, e.g., thyroid [24]. As glioblastoma has a sub-population of stem cells capable of self-renewal and radiotherapy resistance, the response of prominin-1/CD133 to ATRA in U87MG may potentially serve as a GBM cell line model for targeted differentiation strategies.

### 3.2. The Intercellular Adhesion Molecule (ICAM-1; CD54) Is Distributed throughout U87MG with Increased Expression in Response to ATRA

Research suggests that ICAM-1 is potentially an important mediator of tumor migration and invasion in chemotherapy resistant glioblastoma [25]. The findings from this study provide evidence for the presence of ICAM-1/CD54 in the nucleus and cytoplasm of U87MG (Figure 2) and its even distribution throughout the cell culture population. With increasing ATRA concentrations, U87MG show enhanced expression of ICAM-1/CD54 as shown by Western blot analysis. In comparison to non-treated U87MG, exposure to ATRA (10^−6^ M) caused a significant 1.2 ± 4—fold increase (*p* = 0.005) (Figure 2, Appendix A).

When similarly compared against non-treated U87MG, further elevation in ATRA concentration (10^−4^ M) resulted in even higher ICAM-1/CD54 expression levels (3.3 ± 3-fold increase, *p* = 0.0004) (Figure 2). Targeting ICAM-1 may provide a strategy for enhancing the efficacy of anti-angiogenic therapy against GBM and prevent the invasive phenotype of this disease. Hence, U87MG could serve as a model for ICAM-1 manipulation studies and chemotherapeutic testing experimentation.

### 3.3. ATRA Increases Proliferation of U87MG in the Leading Edge in Comparison to the Established Cellular Monolayer

The effects of ATRA on the migration and invasiveness of glioma remains poorly understood, although it is universally accepted that it can induce apoptosis and inhibit proliferation in GBM [26,27,28,29]. Nevertheless, the association between the concentration and effects of ATRA and GBM proliferation remains unclear [30]. U87MG is regarded as a proliferative GBM cell line that forms fast-growing tumors when inoculated orthotopically into a mouse [31]. This study chose to examine the proliferative activity of U87MG (passage 5) following ATRA exposure in two regions of an in vitro culture environment i.e., LE and ECM. Five days after wound formation, exposure to ATRA and EdU labelling, U87MG cells were removed and processed for flow cytometry. Results showed that the ECM and LE cells differed slightly in the forward scatter pattern (Figure 3A) indicative of the presence of mature cells with alternative structural conformation in the former zone. Gating analysis for the selection of cells in G2 phase enabled the effects of plus/minus ATRA exposure to be determined (Figure 3B,C).

The percentage of G2 cells per total events (cell number) was similar in the ECM and LE zones (*p* = 0.21, *n* = 3; Figure 3B,C). These findings revealed that ATRA did not significantly alter the proliferative status of cells grown in the ECM zone when in the presence of 10^−6^ or 10^−^4 M ATRA (*p* = 0.108; *p* = 0.64 respectively) when compared to non-exposed controls. In contrast U87MG in the LE showed significantly increased proliferation upon ATRA exposure (10^−4^ M) in comparison to controls (*p* = 0.028, *n* = 3). This effect was not found at a lower ATRA concentration (10^−6^ M). Results of these experiments, highlight the differential effects of ATRA within an in vitro culture of U87MG. The concentrations of ATRA used in our study did not inhibit proliferation in U87MG or induce apoptosis in contrast to previous findings [30] but may influence late passaged cells (see below). Results show the importance of testing a range of ATRA concentrations and the potential influence of the microenvironment when considering this U87MG cell line in proliferation studies.

### 3.4. Long Non-Coding RNAs (lncRNA) at the Leading Edge Respond to ATRA Treatment in Early Passaged U87MG

Next, this investigation sought to determine whether the lncRNAs *PARTICLE* and *GAS5* are expressed in U87MG given their recognized role in tumor activation and repression respectively [14,32,33]. Dose dependent decreased expression of *PARTICLE* and *GAS5* was identified in the LE of U87MG cultures post exposure to ATRA for 7 days when compared to non-treated controls. Significantly reduced expression of these lncRNAs occurred within the ATRA 10^−7^–10^−4^ M range (Figure 4A,B). When compared to non-treated controls, *PARTICLE* expression showed a significant 22–52% reduction following exposure to ATRA (10^−7^ M, *p* = 0.03; 10^−6^ M, *p* = 0.000019; 10^−5^ M, *p* = 0.000018; 10^−4^ M, *p* = 0.011; Figure 4A). Upon comparison to non-treated controls, *GAS5* expression showed a significant 25–92% lower level after exposure to ATRA (10^−7^ M, *p* = 0.04; 10^−6^ M, *p* = 0.03; 10^−5^ M, *p* = 0.027; 10^−4^ M, *p* = 0.004; Figure 4B). Intriguingly, decreases in *PARTICLE* and *GAS5* were not apparent in the ECM of the culture following ATRA within an identical concentration range and treatment time (Figure 4C,D). Upon passaging, significantly decreased expression of *PARTICLE* (0.4-fold reduction, *p* = 0.029) occurred (Figure 4E), in contrast to *GAS5* which showed elevated levels (0.9-fold increase, *p* = 0.019) when passage #5 was compared to passage #15 following ATRA treatment (10^−4^ M) (Figure 4F). Results highlight the altered expression of lncRNAs within the microenvironment of the U87MG in vitro culture and the influence of passaging on transcription profiles.

### 3.5. Increased Prevalence of Chromosomal Aberrations with Passaging in U87MG Exposed to ATRA

Microscopic evaluation of chromosomal spreads revealed varying susceptibility to aberrations across the 22 autosomes and X gonosome. Frequent chromosomal anomalies included potential inter-chromatid interchanges, interstitial deletions, break discontinuities and dicentric occurrence in U87MG passaged 10–16 times (Figure 5A 1–6). Monosomy was evident at passage 10 with chromosomes 11, 13, 14 and 21 represented as single non-paired chromosomes (Figure 5B). The lack of chromosome pairs became more prevalent by passage 14 onwards (Figure 5C) affecting chromosome 5, 6, 8, 9, 11, 13, 14, 15, 17 and 18. Chromosomes 3, 4, 20, 22 and X were stable up until late passaging (passage 16) was reached by which time polyploidy was evident across the entire karyotype of the U87MG line (Figure 5C). Chromosome 21 was only found as a single chromosome in 70% of chromosomal spreads examined from passage 10 (R.O.I. = 20). Given that U87MG were exposed to ATRA (10^−4^ M) and subsequently passaged, the chromosomal aberrations are potentially the effect of prolonged passaging rather than lengthy ATRA exposure.

## 4. Discussion

The characterization of U87MG (ECACC 89081402) and its response to ATRA were undertaken in this study. Our findings revealed the extensive stem cell attributes of this cell line as shown by the widespread expression of prominin-1 (CD133) which becomes significantly decreased in a dose dependent manner in response to elevated ATRA levels. An intercellular adhesion molecule (ICAM-1, CD54) was also found to be highly expressed in U87MG when exposed to ATRA, offering a potential target against chemotherapy resistance. Focusing on this cell lines response to ATRA enabled the deciphering of alternative proliferation rates between the established cellular monolayer and leading-edge zones. The expression profile of long non-coding RNAs *PARTICLE* and *GAS5* were characterized in this cell line, whereby both responded inversely in a dose dependent manner to serial concentration increases in ATRA, but only at the leading edge. Prolonged U87MG passaging significantly decreased expression of *PARTICLE* in contrast to elevated levels determined for *GAS5* in the presence of ATRA. Karyotyping U87MG confirmed the female origins of this cell line due to the absence of the Y chromosome and presence of the X chromosomal pair. The frequency of chromosomal aberrations increased with passaging from passage 10 to 16 with extensive polyploidy evident at the late stage. This report reveals the dynamic nature of this cell line that provides challenges for stable data acquisition for glioblastoma research. U87MG (ECACC 89081402) and U87MG (HTB-14; ATCC) share identical STR profiles except for sex chromosome profiles according to supplier websites. These sources indicate that U87MG (ECACC 89081402) reaches 100% confluency between seventy to ninety hours cultivation, in contrast to U87MG (HTB-14; ATCC) which tends to become non-adherent during proliferation.

Prominent features of glioblastoma which hamper effective treatment include rapid growth and clonal spatial and temporal heterogeneity with stem-like features [34]. Prominin-1 (CD133) is regarded as a tool for hematopoietic stem cell isolation [35] as well as an identification marker for targeting populations of malignant transformation in certain cancers such as leukemia [35]. The extensive and high levels of prominin-1 in U87MG reported here is supported by others utilizing novel adherent culture methods [36]. ICAM-1 was also prevalent in U87MG, which is associated with invasion and metastases in several types of cancer [25]. It has been shown that inhibition of ICAM-1 reduced glioma invasion in vitro and in vivo [37]. Targeting ICAM-1 offers potential for overcoming glioblastoma’s resistance to antiangiogenic therapy and improve prognosis [25].

Compromised retinoid signaling in carcinogenesis proposed a role for retinoic acid deficiency in tumor development [38]. ATRA was shown to inhibit proliferation of some glioma cell lines [39]. However, the therapeutic role of retinoic acid in cancer remains uncertain due to increased risks of mortality associated with its supplementation [7]. It has been proposed that gliomas may be initiated by an increase in endogenous production of retinoic acid in glia with alternative effects on receptor expression levels [7]. The differential effects of ATRA on the leading edge versus the established cellular monolayer within an in vitro culture dish shown in this report, highlights the ambiguities of its proliferative instigation properties.

Evidence has shown the important role of lncRNAs in tumor suppressor regulation [7,13,14,15,16,17,40] yet their involvement in GBM is very much under explored to date [41]. The lncRNA *PARTICLE* controls the expression of tumor suppressors *MAT2A* and *WWOX* in breast cancer and osteosarcoma respectively [13,14,15,16,17]. This foremost study represents the initial demonstration that *PARTICLE* is expressed in U87MG with significantly higher levels of expression at P5 compared to P15 (0.4-fold decrease, *p* < 0.05). These results support the role of *PARTICLE* as a negative regulator of tumor suppressors [13,14,15,16,17]. Nevertheless, the potential for the role of *PARTICLE* in epigenetic modulation associated with tumorigenicity loss can also be possible [13]. Findings reveal its significant dose dependent reduction to elevated ATRA concentrations in leading edge U87MG in vitro at the P5 stage. Surprisingly, this effect was not found in the established cellular monolayer zone, highlighting the differential response of lncRNA to this retinoid within the in vitro microenvironment. The implications for this are so far unknown and may reflect the dysregulated expression of lncRNAs associated within brain tumors [42,43].

LncRNA profiling of GBMs using The Cancer Genome Atlas database, identified *GAS5* among others closely associated with patient overall survival [44]. A functional role as a tumor suppressor has been previously reported for *GAS5* in U87 (obtained from the Shanghai Institutes for Biological Sciences Cell Resource Center) [40]. Our findings showed a significant elevation in *GAS5* with passaging as reflected in the 4.8-fold increase in expression levels in P15 compared to P5 in response to similar concentrations of ATRA. Likewise, findings showed an inverse expression level of *GAS5* with increased ATRA levels only in leading-edge cells and not in the established cellular monolayer zone similar to the *PARTICLE* profile in this regard.

Continuous serial in vitro cellular passaging is routinely undertaken in biological research laboratories. It has been reported that late passage U87MG show decreased tumorigenicity with lower invasion capabilities than early passaged cells without significant differences in proliferation and migration [41]. Given the suggestion that passage-induced changes may lead to notable changes in biological characteristics [41], this study examined the potential for chromosomal alterations post ATRA exposure between passage 10 and passage 16 of U87MG. While other studies have investigated the effects of serial passaging up to 100 times, our study noted the extensive polyploidy by passage 16 followed by cellular deterioration and apoptosis. This work supports the necessity to avoid lengthy serial passaging and encourages the use of identical or similar passage number for experimental data acquisition. Human U87MG is one of the most commonly studied grade IV glioblastoma cell lines. It has come into focus recently given the probable misidentification of U87MG procured from ATCC [12]. Genomic sequencing of U87MG unveiled an exceptionally dense mutational landscape that supports a model where cancer mutations occur via structural instability rather than novel point mutations [45].

In conclusion, U87MG (ECACC 89081402) offers a glioblastoma cell line model for exploitation of stem cell differentiation and chemotherapy resistance. Caution is advised when assessing proliferation outputs due to molecular differences within the in vitro microenvironment. An avoidance of continuous passaging is recommended due to the development of karyotype aberrations. While more representative GBM models are keenly sought, the U87MG may still offer a platform for glioblastoma research on condition that the challenges associated with this cell line do not go unheeded by the scientific community.

## Figures and Tables

**Figure 1 brainsci-11-00812-f001:**
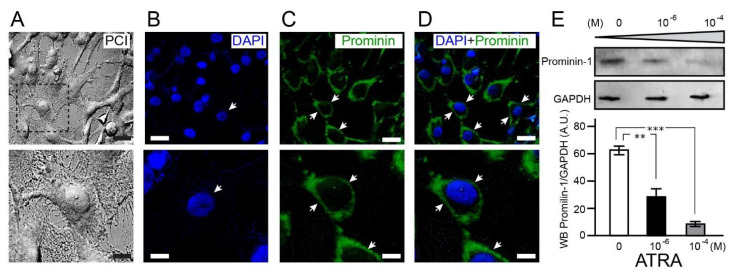
The widely expressed stem cell marker prominin-1 in U87MG is curtailed in a dose dependent manner by ATRA. Phase contrast image (PCI) of early passaged U87MG grown to 8090% confluency ((**A**), upper). Boxed region shown in higher magnification ((**A**), lower) showing a representative U87MG cell with large nucleus and cytoplasmic projections. Confocal micrographs with U87MG nuclear staining (arrow; DAPI—Blue; (**B**)), stem cell marker prominin-1 revealing strong cytoplasmic expression pattern (arrow; Alexa fluor 488; Green; (**C**)) without ATRA treatment. Merged images (**D**) showing lack of prominin-1 peri-nuclear expression. Scale bars: 20 µm (upper) and 10 µm (lower). Representative Western blot highlighting decreased prominin-1 expression in U87MG with increasing ATRA exposure and GAPDH as a loading control ((**E**), upper). Histograms depicting Western blot analysis and significant inverse relationship between prominin-1 expression and ATRA exposure ((**E**), lower). ** *p* < 0.005, *** *p* < 0.0005, *n* = 2.

**Figure 2 brainsci-11-00812-f002:**
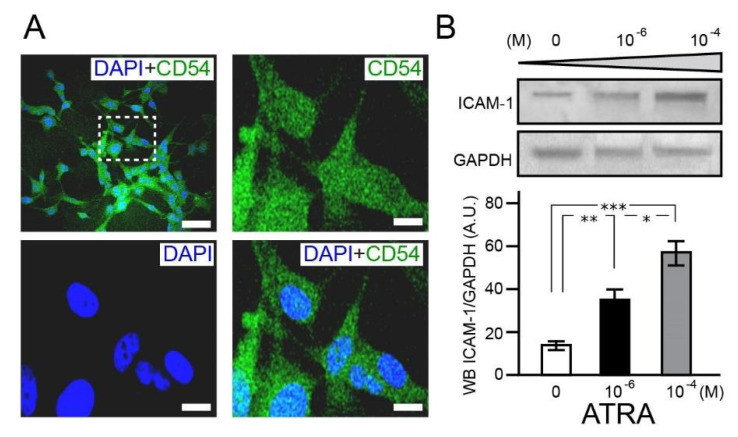
Intercellular adhesion molecule 1 is located in the nucleus and cytosol with increased expression in response to ATRA. Confocal micrographs of U87MG showing immunohistochemical detection of ICAM-1 (CD54; Alexa fluor 488; green) and nuclear stained DAPI (blue) (**A**) after 7 days of ATRA (10^−4^ M) treatment. Scale bar 100 µm. White dashed box region ((**A**), upper left) is shown in higher magnification (×40, (**A**)) to highlight the presence of ICAM-1 in the nucleus and cytosol. Scale bar 20 µm. Representative Western blot (upper (**B**)) and analysis (lower (**B**)) indicates the incremental increased expression of ICAM-1 in response to higher ATRA concentrations. * *p* < 0.05, ** *p* < 0.005, *** *p* < 0.0005, *n* = 2.

**Figure 3 brainsci-11-00812-f003:**
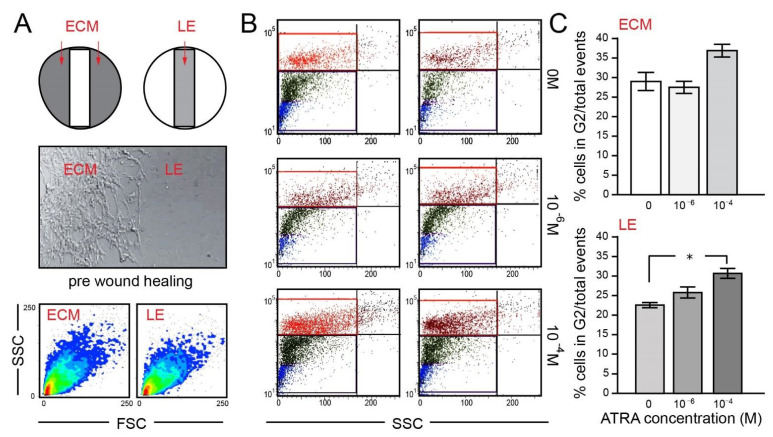
ATRA influences U87MG proliferation only in the leading edge. Schematic of the location of the established cellular monolayer (ECM) and leading edge (LE) zones in a tissue culture in vitro environment post wound formation (**A**); upper). Confocal micrograph of U87MG cells in the ECM and LE before would healing (**A**); lower). Dot blot density cytogram of side scatter (SSC) versus forward scatter (FSC) of U87MG in the ECM (left) and LE (right), providing an indication of the size and granularity of cells respectively. The red/yellow/green/blue hot spots represent increasing numbers of events resulting from discrete cell populations ((**A**); bottom). Side scatter plot of Click-iT Plus EdU AlexaTM 647 fluorescence incorporated cells in ECM and LE zones plus/minus ATRA with gating to demarcate the population in G2 growth phase (upper left (UL); red). Populations in G1 and S phase are indicated in blue and green respectively (**B**). Histograms showing the percentage of cells in G2 per total events (assessed cell population) indicating significance of ATRA (10^−4^ M) in U87MG cells in the LE zone. * *p* < 0.05, *n* = 3, (**C**).

**Figure 4 brainsci-11-00812-f004:**
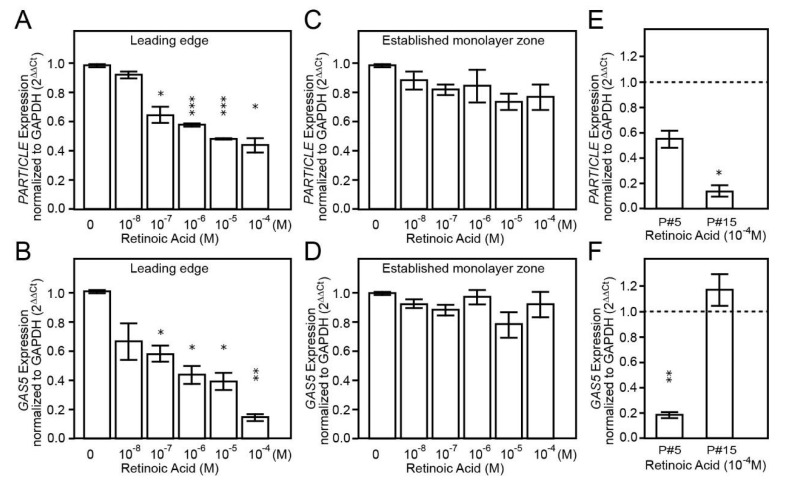
Long non-coding RNA *PARTICLE* and *GAS5* respond to ATRA in U87MG leading edge cells and not in the established cellular monolayer zone with differential passage effects. Histograms of quantitative real time PCR data of *PARTICLE* and *GAS5* showing decreased expression profiles in leading edge cells in response to increased ATRA concentrations (**A**,**B**). Neither *PARTICLE* nor *GAS5* show differential expression in response to increased ATRA concentrations in the established cellular monolayer zone of U87MG (**C**,**D**). Histograms highlight reduced levels of *PARTICLE* with passaging (**E**) in contrast to *GAS5* which shows significantly elevated expression in later passages (**F**). * *p* < 0.05, ** *p* < 0.005, *** *p* < 0.0005, *n* = 3.

**Figure 5 brainsci-11-00812-f005:**
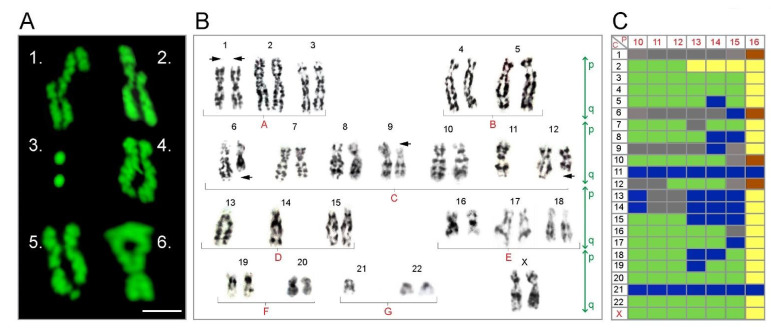
Chromosomal aberrations with passaging in U87MG exposed to ATRA. Representative examples of pseudo-colored chromosomes after Giemsa staining showing various chromosome aberrations in U87MG cells post passage 10–16. Chromosome 6 alignment with chromosome 13 with potential inter-chromatid interchanges (**A1**), chromosome 3 with chromatid p arm deletion (**A2**), interstitial deletions (**A3**), chromosome 12 with chromatid protrusion (**A4**), chromosome 4 with potential q-arm breakage (**A5**), chromosome 6 telomeric alignment with chromosome 18 or dicentric chromosome (**A6**). Representative karyotype spread of Wright stained chromosomes from U87MG passage 10 cells. Monosomy indicated by absence of chromosome. P and q represent short and long chromosomal arms. Chromosomal groups represented by letters A–G, and chromosomes numbered according to standard practice for a female karyotype. Arrows point to deletions (black arrows) (**B**). Illustrative table summarizing chromosomal anomalies arising with passaging U87MG. Passage (P) number shown across the top, chromosome (**C**) number shown along the side of the table. Colors represent chromosomal deletions (grey), stability (green), anomalies (red), polyploidy (yellow) and monosomy (navy blue).

## Data Availability

All material and data will be available upon request.

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
