# Peer review of "All-Trans Retinoic Acid Fosters the Multifarious U87MG Cell Line as a Model of Glioblastoma"

_brainsci, 2021, doi:10.3390/brainsci11060812_

Round 1

Reviewer 1 Report

  1. While the authors note that there are problems with the ATCC version of U87MG cell line, including the presence of a Y chromosome, they should discuss further how the results they have obtain in their characterization of the ECACC stock differ from the ATCC stock.  If experiments performed in parallel were available that would be ideal, but if not, at least a summary of how the experimental results would have been expected to differ from the ATCC version of the cells.
  2. Loss of in vivo tumorigenicity is a phenomenon generally associated with culturing tumor cell lines in vitro, and this is can be due to reversible epigenetic changes rather than more dramatic genetic alterations.  While experiments testing passage of tumors in vivo (from animal to animal) go beyond the scope of this report, the potential for epigenetic modulation assocaited with loss of tumorigenicity should be mentioned.   
  3. Some of the sentence structures should be improved for clarity (e.g., 60, 426, 430)   

Reviewer 2 Report

This study aimed to investigate the proliferative characteristics, gene expression profile and karyotype of U87MG (ECACC 89081402) post ATRA exposure to highlight subtle challenges associated with this widely utilised laboratory cell line.

The manuscript is well written and the experiments are well described as well as results and discussion. However some revision are required:

1- All Figures in the text are in low quality, author must include high quality Figures to be clear when zoomed.

2- Images of control cells without ATRA exposure are required in both Figure 1 (p6) and Figure 2 (p7).

3- The author investigate the proliferative characteristics, gene expression profile and karyotype of U87MG (ECACC 89081402) post ATRA exposure. The proliferative characteristics and gene expression profile of U87MG must be studied also when cells are incubated under stressful conditions (under anticancer drug exposure such paclitaxel) to make difference between cells expression under different incubation conditions (without treatment, ATRA exposure and anticancer exposure).  

Round 2

Reviewer 2 Report

Despite the modifications given by the authors, some additional corrections are required:

- Phase contrast images of control and treated cells are required in Figure 1.

Likewise Control and ATRA treated cells of all confocal micrographs with U87MG nuclear staining DAPI Blue and Alexa Fluor 488 Green must be represented.

Question: How can the authors pointed the lack of Prominin-1 in the Merged images without control?

- Figure 2: the same suggestions control and treated cells with ATRA must be represented in the Figure.

- Figure 5:  the authors described the chromosomal aberrations but only the treated cells are represented. Chromosomes of control cells without aberration are required.
